# **TweetNERD** - End to End Entity Linking Benchmark for Tweets

**Shubhanshu Mishra**[*]   **Aman Saini**   **Raheleh Makki**   **Sneha Mehta**   **Aria Haghighi**

**Ali Mollahosseini**
Twitter, Inc.
{smishra,amansaini,rmakki,snehamehta}@twitter.com
{ahaghighi,amollahosseini}@twitter.com

## Abstract

Named Entity Recognition and Disambiguation (NERD) systems are foundational for information retrieval, question answering, event detection, and other natural language processing (NLP) applications. We introduce TweetNERD, a dataset of 340K+ Tweets across 2010-2021, for benchmarking NERD systems on Tweets. This is the largest and most temporally diverse open sourced dataset benchmark for NERD on Tweets and can be used to facilitate research in this area. We describe evaluation setup with TweetNERD for three NERD tasks: Named Entity Recognition (NER), Entity Linking with True Spans (EL), and End to End Entity Linking (End2End); and provide performance of existing publicly available methods on specific TweetNERD splits. TweetNERD is available at: https://doi.org/10.5281/zenodo.6617192 under Creative Commons Attribution 4.0 International (CC BY 4.0) license [Mishra et al., 2022]. Check out more details at https://github.com/twitter-research/TweetNERD.

## 1   Introduction

Named Entity Recognition and Disambiguation (NERD) [Mihalcea and Csomai, 2007, Cucerzan, 2007, Derczynski et al., 2015, Kulkarni et al., 2009] is the task of identifying important mentions or Named Entities in the text and linking those mentions to corresponding entities in an underlying Knowledge Base (KB). The KB can be any public knowledge repository like Wikipedia or a custom knowledge graph specific to the domain. NERD for social media text [Derczynski et al., 2015, Mishra and Diesner, 2016, Mishra, 2019], in particular Tweets is challenging because of the short textual context owing to the 280 character limit of Tweets. There exist multiple datasets [Derczynski et al., 2015, Mishra, 2019, Dredze et al., 2016, Derczynski et al., 2016, Spina et al., 2012, Rizzo et al., 2016, Yang and Chang, 2015, Fang and Chang, 2014, Locke, 2009, Meij et al., 2012, Gorrell et al., 2015] for developing and evaluating NERD methods on Tweets. However, these datasets have limited set of Tweets, are temporally biased (i.e. Tweets are from a short time period, more details in section C.1), or are no longer valid because of deleted Tweets (see Table 3). In this work, we introduce a new dataset called TweetNERD which consists of 340K+ Tweets annotated with entity mentions and linked to entities in Wikidata (a large scale multilingual publicly editable KB). TweetNERD addresses the issues in existing NERD datasets for Tweets by including Tweets from a broader time window, applying consistent annotations, and including the largest collection of annotated Tweets for NERD tasks. Figure 1 compares TweetNERD with existing Tweet entity linking datasets, proving its increases scale. Furthermore, we describe two splits of the dataset which we use for evaluation. These splits called TweetNERD-OOD and TweetNERD-Academic

---

[*]Corresponding Author

36th Conference on Neural Information Processing Systems (NeurIPS 2022) Track on Datasets and Benchmarks.

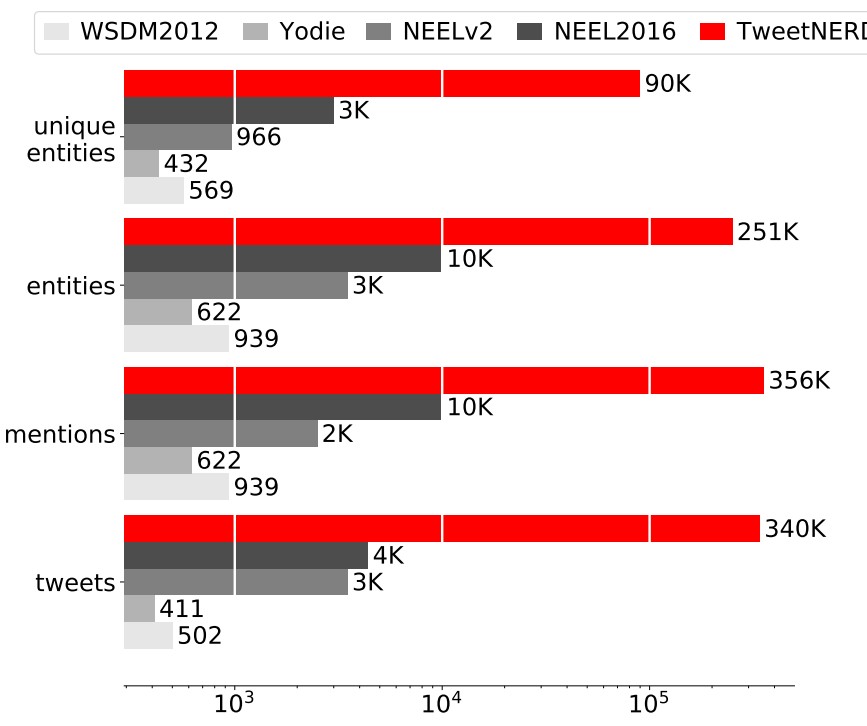

Figure 1: Comparison with existing Tweet entity linking datasets

allow assessing out of domain (OOD) and temporal generalization respectively. `TweetNERD-OOD` split consists of Tweets in a shorter time frame that are over-sampled with harder to disambiguate entities. It is useful to assess out of domain performance. Conversely, `TweetNERD-Academic` split is a temporally diverse dataset of non-deleted Tweets from a collection of existing academic benchmarks that have been re-annotated with the new annotation guidelines. `TweetNERD` has already been used by Hebert et al. [2022] for evaluating dense retrieval for candidate generation in presence of noisy NER spans. `TweetNERD` should also foster research in better utilization of social graph context of Tweets [Kulkarni et al., 2021, Li et al., 2022] in improving NERD task performance, and assessment of bias in NERD systems [Mishra et al., 2020]. `TweetNERD` is available at: https://doi.org/10.5281/zenodo.6617192 under Creative Commons Attribution 4.0 International (CC BY 4.0) license [Mishra et al., 2022]. Check out more details at https://github.com/twitter-research/TweetNERD.

## 1.1 Related works

Named Entity Recognition and Disambiguation (NERD) is a prominent information extraction task. There exist multiple datasets [Derczynski et al., 2015, Mishra, 2019, Dredze et al., 2016, Derczynski et al., 2016, Spina et al., 2012, Rizzo et al., 2016, Yang and Chang, 2015, Fang and Chang, 2014, Locke, 2009, Meij et al., 2012, Gorrell et al., 2015, Eskander et al., 2022] for Named Entity Recognition (NER), NERD, Cross Domain Co-reference Retreival (CDCR), or Entity Relevance. Most datasets have Tweets sampled from a given time-period, which were either annotated for NER or NERD. The annotation varied by linked knowledge base being either DBPedia [Gorrell et al., 2015], Wikipedia [Rizzo et al., 2016], or Freebase [Fang and Chang, 2014]. Our work closely follows the annotation process of Gorrell et al. [2015] of linking entities using a crowd sourcing platform and doing both NER and Entity Disambiguation tasks. Our data collection process differs in terms of sampling Tweets from a diverse temporal window and the inclusion of more diverse set of entities (see section 4.1).

## 2 Terminology

We use the following terminology throughout the rest of the paper: (1) **knowledge base (KB)**: Underlying knowledge base of entities, we use Wikidata [Vrandečić and Krötzsch, 2014]. (2) **document id** ($id$): Id of the document with entities, and optional meta-data e.g. date; (3) **mention** ($m$): a phrase in document $d$ identified by a start offset $s$ and end offset $e$; (4) **start** ($s$): starting offset of mention $m$. The offset is dependent on the encoding of the data (TweetNERD uses byte offsets for the text encoded using utf-16-be); (5) **end** ($e$): ending offset of mention $m$ in the same format as ($s$), such that $len(m) = e - s$; (6) **NIL**: If a mention can't be linked to any entity in KB; (7) **entity id** ($eid$): Linked entity Id in KB or NIL; and (8) **candidate set** ($C$): Possible candidates for $m$ in KB and NIL.

## 3 Annotation Setup

**Annotators**   We leveraged a team of trained in-house annotators who utilized a custom annotation interface to annotate the Tweets. A pool of annotators was trained with detailed labeling guidelines and multiple rounds of training iterations before actually starting to annotate the Tweets in TweetNERD. The guidelines included examples of Tweets with linked entities, and instructions on how to disambiguate between potential candidates using the Tweet context, media, time and other factors. A much simplified version of the interface is shown for the purpose of illustration (see Figure 2). The annotators were required to pass a qualification quiz demonstrating their understanding of the task to be eligible as an annotator.

**Annotation Task**   The annotation task required identifying all mentions $m$ in a Tweet and assigning a Wikidata ID, $eid$, for each $m$. The annotators had to highlight the mention and then use Wikidata search interface to find the correct $eid$ (e.g. $m$=Twitter and $eid$=Q918). Annotators could edit the search phrase to differ from $m$ to correct for spelling errors, or expand it with additional words in order to find a suitable entity. If there is no valid Wikidata ID for $m$, annotators assigned $eid$=NOT FOUND. If annotators thought that the Tweet context is not clear enough to disambiguate between the returned candidates they assigned $eid$=AMBIGUOUS. The Wikidata ID for a given Wikipedia page is obtained by clicking on the Wikidata item link located on the left panel of the Wikipedia page. TweetNERD annotation was done in batches where around 25K Tweet ids for each batch were sampled via the setup described in the next section. We annotated a total of 14 batches for the TweetNERD dataset.

**Eligible Mentions**   Annotators were instructed to select mentions $m$ in a Tweet which refer to the longest phrase corresponding to a named entity that can be identified as a Person, Organization, Place etc. (see Table 1 for full list and details). A mention can also be contained within a hashtag if it corresponds to an entity e.g. #FIFA.

**Correct Candidate**   Annotators were instructed to prefer an $eid$ which is likely to have a Wikipedia page. The most appropriate $eid$ could depend on the following: (a) full text of the Tweet, (b) the URL or media attached to the Tweet, (c) the temporal context of the Tweet (annotators can search for $m$ on Twitter around the same date as the Tweet), (d) the Tweet thread it is part of (i.e. which Tweet it is replying to and the list of Tweets which replied to it) (e) the user of the Tweet.

**Annotation Aggregation**   Each Tweet was annotated by *three* annotators and $(m, eid)$ pairs that were selected by *at least two* annotators were considered **gold annotations**. We include all annotations (including non-gold) as part of the final dataset to support additional analysis (e.g. studying annotation noise).

**Difficulty of the Annotation Task**   Entity Linking is inherently a difficult task due to name variations (multiple surface forms for the same entity) and entity ambiguity (multiple entities for the same mention) [Shen et al., 2014]. In addition, based on the type of application and the coverage of the underlying knowledge base this task can become challenging even for humans. E.g. we asked the annotators to link a mention to the most specific entity in the knowledge base (i.e. Wikidata), this assumption forces all other candidate entities (even if close) for that mention as incorrect. For instance, if a Tweet is about the Academy Awards this year (2022), we only consider Q66707597 (94th Academy Awards) as the correct entity and not Q19020 (Academy Awards), while Q19020 is

| | |
|---|---|
| **Id=1**: I love [**Twitter**][ENTITY] | |
| Candidates: **Q918**, NOT FOUND, AMBIGUOUS | |

**Id=1**: I love [**Twitter**]_[ENTITY]_
Candidates: **Q918**, NOT FOUND, AMBIGUOUS

**Id=2**: [**Paris**]_[ENTITY]_ is regarded as the world's fashion capital
Candidates: **Q90**, Q79917, NOT FOUND, AMBIGUOUS

**Id=3**: [**Anil**]_[ENTITY]_ is playing
Candidates: NOT FOUND, **AMBIGUOUS**

Figure 2: **Simplified version of the annotation interface.** Selected mentions and entities are in **Bold**. Important thing to note is that the annotators are shown only the Tweet text. They use the functionality provided in the interface to query the eligible knowledge base candidates. Each annotator can select multiple mentions in a Tweet but link each mention ($m$) to only a single Entity Id ($eid$).

Table 1: Example of types of entities to identify in the text

| Type | Examples |
|---|---|
| Person | Politicians, sports players, artists, celebrities, fictional characters, scientists, singers, musicians, journalists, social media celebrities, and others
Examples: Kanye West, Sachin Tendulkar, Donald Trump, Harry Potter, Jon Snow |
| Place | Countries, Cities, Monuments, Parks, rivers, and others
Examples: Paris, Nigeria, Statue of Liberty |
| Organization | Companies, governments, NGOs, social movements, music bands, sports teams, social organizations, volunteer organizations, and others
Examples: Backstreet Boys, Los Angeles Lakers, Black Lives Matter |
| Products | Websites, Softwares, applications, video games, technology gadgets, devices, and others
Examples: PlayStation, iPhone, GoFundMe, Roblox |
| Works of Art | Movies, Albums, Books, Comics, Video Games, TV Shows, Social Media videos, and others
Examples: Friends, The Office, Lupin |
| Scientific Concepts | Names of diseases, drugs, names of algorithms, scientific methods and techniques, scientific names of organisms, names of disasters, and others
Examples: COVID-19, SARS-COV19, Hurricane Katrina, Cyclone Idai |

the correct entity if the Tweet is about Academy Awards in general. While this allows for temporally sensitive annotations, it makes the task difficult compared to most classification tasks, leading to a negative impact on inter-annotator agreement (see discussion in section 4.4).

## 4 Tweet End To End Entity Linking Dataset

### 4.1 Sampling

`TweetNERD` consists of English Tweets most of which were created between Jan 2020 and Dec 2021. Tweet language was identified using the Twitter Public API endpoint. Additionally, we discarded Tweets which were NSFW[2], too short ($\leq 10$ space separated tokens), and included $\geq 2$ URLs or $\geq 2$ user mentions or $\geq 3$ Hashtags. Since the dataset was annotated in batches, we were able to improve our sampling technique with each batch. Our initial approach of upsampling Tweets with high retweets and likes (Tweet-actions) resulted in a large proportion of Tweets with empty annotations. To mitigate this, we experimented with approaches which select Tweets that are more likely to have an entity. Some of these approaches included: (a) using in-house NER models [Mishra et al., 2020, Eskander et al., 2020] to check for NER mentions, (b) using phrase matching techniques [Mishra and Diesner, 2016] to match phrases from Tweet text with the Wikidata entity titles, (c) sampling

---

[2]NSFW - Not Safe For Work

based on phrase entropy to detect difficult phrases (described in the next paragraph), (d) overall Tweet favorite based sampling, and (e) search page click based sampling. Within each approach, we perform a stratified sampling to select Tweets equally from each sampling bucket. The full dataset `TweetNERD` is comprised of different proportions of each of these buckets.

**Entropy based sampling**   We wanted to include tweets containing phrases representing a diverse set of wikidata entities in terms of entity popularity as well as disambiguation difficulty. We used aggregate wikipedia page views ($eid_{views}$) across all language pages of a wikidata entity as a proxy for its popularity. Then the phrase entropy was defined as $H = \sum p * log(p)$ using the probability $p = p(eid|m) = eid_{views} / \sum eid_{views}$. Each phrase is then classified as one of the high, medium, or low entropy phrase using the entropy score distribution. Finally, we sample an equal number of Tweets from each phrase entropy bucket.

## 4.2   Data Splits

While `TweetNERD` consistes of 340K+ Tweets, we highlight two explicit data splits of `TweetNERD`, namely `TweetNERD-OOD` and `TweetNERD-Academic`, which have been used as test sets for evaluation in this paper. The purpose of these two splits is to measure out of domain performance and temporal generalization respectively.

**`TweetNERD-OOD`**   It is a subset of 25K Tweets used for evaluating existing named entity recognition and linking models. `TweetNERD-OOD` is sampled in equal proportion based on the entropy of the contained NER mentions. Mentions with few, less diverse candidates fall in the low entropy buckets whereas mentions with many, high diversity candidates fall into the high entropy buckets. We first sample Tweets into high, medium and low entropy mention buckets, and then perform stratified sampling based on Tweet actions to divide these buckets into sub-buckets. This approach helps us to evaluate all models against a variety of Tweets with varying levels of difficulty and popularity.

**`TweetNERD-Academic`**   It is a subset of 30K Tweets to benchmark entity linking systems on Tweets already sampled in existing academic benchmarks (mostly from [Derczynski et al., 2015, Mishra, 2019]). We identify all the Tweet ids across existing NERD, NER, NED, and syntatic NLP task datasets for Tweets and hydrate these ids using the Public Twitter API. We ended up with 30,119 Tweets across these datasets which are still available (see Table 3). Its important to note that these Tweets were annotated again using our latest annotation setup to comply with the `TweetNERD` guidelines. Our intention for including this split is to add a layer of temporally diverse and already benchmarked datasets.

**Re-annotation of academic benchmarks in `TweetNERD-Academic`**   We re-annotate the academic benchmark datasets in `TweetNERD-Academic` using our guidelines and setup to ensure consistency with the rest of our dataset. This choice was made as opposed to including the existing annotations from these datasets for the following reasons. First, not all of these datasets are annotated for the end to end NERD task, i.e. some only have NER and some only have NED annotations. Second, the knowledge base used for each NERD annotation is not Wikidata. Instead, some datasets link to DBpedia, some to English Wikipedia. Third, the notion of entities to annotate varies across the datasets and would require a lot of reconciliation to make a consistent benchmark dataset, e.g. Rizzo et al. [2016] annotates Hashtags and user mentions as entities but `TweetNERD` does not allow mentions to be tagged as entities. Finally, many of the Tweets (20-40%, see table 3) from these datasets are not available via the public API, however, those which are still available are likely to be available for a longer duration which makes this benchmark more stable. We show some examples of annotations in `TweetNERD-Academic` versus existing academic benchmarks in table 2. Detailed description of each of these datasets is provided in section C.1. Finally, we observed high overlap between `TweetNERD-Academic` and academic datasets. E.g. using Yodie as the closest academic dataset in terms of our annotation guidelines, we found that `TweetNERD-Academic` matches 77% Yodie mention level annotations as well as 87% mention annotations at the Tweet level. At the mention-entity level `TweetNERD-Academic` matches 65% Yodie annotations and 80% at the Tweet level (we map DBPedia entity annotations in Yodie to their Wikidata ID).

**Flexibility for Further Analysis**   As seen above, we have identified two subsets of the dataset (`TweetNERD-OOD` and `TweetNERD-Academic`) which we use as test sets for evaluation in this

Table 2: Annotations in `TweetNERD-Academic` versus annotations in existing benchmarks.

| |
|---|
| **Text**: Press release:"Will England fans be hit by penalties on their next energy bill?" Please make it stop. **Yodie**: England (Dbp:England); **TweetNERD**: England (Q21) |
| **Text**: #DMG #GILDEMEISTER presents the new GILDEMEISTER energy monitor, read more at [URL]. **Yodie**: GILDEMEISTER (6, 18, Dbp:Gildemeister_AG), GILDE-MEISTER (36, 48, Dbp:Gildemeister_AG); **TweetNERD**: GILDEMEISTER (6, 18, Q100151808), GILDEMEISTER (36, 48, Q100151808) |
| **Text**: Wiz Khalifa went suit shopping with Max Headroom. #grammys #80s [URL]. **TGX**: Max Headroom (NA, NA, NA); **TweetNERD**: Wiz Khalifa (0, 11, Q117139), Max Headroom (36, 48, Q1912691) |

paper. While these two datasets can be used for standard benchmarking for tasks similar to those presented in this paper, we would like to emphasize the flexibility of `TweetNERD` in evaluating a wide range of tasks. For example, one could split the full `TweetNERD` dataset temporally to test existing models for temporal generalization or one could split `TweetNERD` based on seen and unseen mentions and entities to assess robustness. `TweetNERD` can also be randomly split into train, validation, and test splits that can be used to evaluate in-domain performance of models. To align ourselves with the traditional machine learning benchmark formats, we also provide canonical train, validation, and test splits of the data created by extracting random samples of 25K tweets for test and 5K for validation from `TweetNERD` excluding `TweetNERD-OOD` and `TweetNERD-Academic`. While we do not report any results on this test split in this paper, we encourage researchers to use these splits along with `TweetNERD-OOD` and `TweetNERD-Academic` to ensure reproducibility.

**Adapting to Temporal Dynamics of Knowledge Bases**   Knowledge Bases are dynamic and new entities are added with time and since NERD datasets are not updated with time there might be discrepancies in model evaluation with reference to a static NERD test set. This is a common limitation of Entity linking evaluation. In `TweetNERD` this would only affect the NIL predictions as opposed to linking predictions. An entity which in 2014 was marked as NIL (because of absence from Wikidata) may be marked correctly now. This can be addressed easily by factoring in the creation date of the entity in Wikidata. This way any entity whose creation date in Wikidata is after the Tweet date can be marked as NIL. This can allow for temporal evaluation.

### 4.3   Data Statistics

Table 3: Details of `TweetNERD-Academic` (same Tweet could occur in multiple datasets).

| dataset | Tasks | Total Tweets | Found | Found % |
|---|---|---|---|---|
| **Tgx** [Dredze et al., 2016] | CDCR | 15,313 | 9,790 | 63.9 |
| **Broad** [Derczynski et al., 2016] | NER | 8,633 | 6,913 | 80.1 |
| **Entity Profiling** [Spina et al., 2012] | NER | 9,235 | 6,352 | 68.8 |
| **NEEL 2016** [Rizzo et al., 2016] | NERD | 9,289 | 2,336 | 25.1 |
| **NEEL v2** [Yang and Chang, 2015] | NERD | 3,503 | 2,089 | 59.6 |
| **Fang and Chang [2014]** | NERD | 2,419 | 1,662 | 68.7 |
| **Twitter NEED** [Locke, 2009] | NERD & IR | 2,501 | 1,549 | 61.9 |
| **Ark POS** [Gimpel et al., 2011] | POS | 2,374 | 1,313 | 55.3 |
| **WikiD** | NED | 1,000 | 504 | 50.4 |
| **WSDM2012** [Meij et al., 2012] | Relevance | 502 | 415 | 82.7 |
| **Yodie** [Gorrell et al., 2015] | NERD | 411 | 288 | 70.1 |

**TweetNERD.**   `TweetNERD` consists of 340K unique Tweets that collectively contain a total of 356K mentions that are linked to 90K unique entities. Of the 356K mentions, 251K are linked to non-NIL entities, and 104K to NIL entities. As can be observed in Figure 1, `TweetNERD` is the largest data set compared to existing benchmark datasets for Tweet entity linking. More details about the salient mentions, entities, and mention-entity pairs in `TweetNERD` can be found in Table 4.

Table 4: Salient entities, mentions, and mention-entity pairs in `TweetNERD` full dataset and subset. Entity refers to $eid$ - the linked Wikidata ID, Mention refers to $m$ - the annotated phrase in the Tweet, and Mention-Entity refers to $(m, eid)$ - a unique tuple of <mention, entity>.

---

**Full data set**

---

**Mention Entity**: Total: 356345, Unique: 166379
Head: "'grammys' <Q630124>" (6272), "'mark lee' <Q26689986>" (2341), "'aria' <AMBIGU-OUS>" (2103), "'whatsapp' <Q1049511>" (1521), "'isabella' <AMBIGUOUS>" (1260)
Mid: "'david mabuza' <Q1174142>" (2), "'neha sharma' <Q863745>" (2)
Tail: "'ian darke' <Q5981359>" (1), "'antony perumbavoor' <Q55604079>" (1), "'sansone' <NOT FOUND>" (1), "'prairie state college' <NOT FOUND>" (1), "'konga' <NOT FOUND>" (1)

---

**Mention**: Total: 356345, Unique: 143762
Head: 'grammys' (7059), 'aria' (2461), 'mark lee' (2342), 'whatsapp' (1602), 'isabella' (1471)
Mid: 'nam joo hyuk' (2), 'sharpsburg' (2)
Tail: 'iain banks' (1), 'michael odewale' (1), 'chlorine cougs' (1), 'rock your baby' (1), 'georgia dome' (1)

---

**Entity**: Total: 356345, Unique: 90938
Head: 'NOT FOUND' (59704), 'AMBIGUOUS' (44752), 'Q630124' (7886), 'Q26689986' (2364), 'Q108112350' (2094)
Mid: 'Q9196194' (2), 'Q331613' (2)
Tail: 'Q107362802' (1), 'Q81101633' (1), 'Q17361809' (1), 'Q1177' (1), 'Q741395' (1)

---

**Without `TweetNERD-Academic`**

---

**Mention Entity**: Total: 312581, Unique: 159468
Head: "'mark lee' <Q26689986>" (2341), "'aria' <AMBIGUOUS>" (2103), "'whatsapp' <Q1049511>" (1521), "'isabella' <AMBIGUOUS>" (1260), "'tajin' <Q3376620>" (1016)
Mid: "'cannes2021' <Q42369>" (2), "'zeynep' <NOT FOUND>" (2)
Tail: "'slave play' <Q69387965>" (1), "'Prada' <Q193136>" (1), "'gansu' <Q42392>" (1), "'iowa state capitol' <Q2977124>" (1), "'konga' <NOT FOUND>" (1)

---

**Mention**: Total: 312581, Unique: 137782
Head: 'aria' (2461), 'mark lee' (2342), 'whatsapp' (1602), 'isabella' (1471), 'matilda' (1434)
Mid: 'jamelia' (2), 'mohammad rafi' (2)
Tail: 'petr yan' (1), 'wooiyik' (1), 'billie dove' (1), 'bucks fizz' (1), 'georgia dome' (1)

---

**Entity**: Total: 312581, Unique: 87430
Head: 'NOT FOUND' (58678), 'AMBIGUOUS' (44202), 'Q26689986' (2364), 'Q108112350' (2094), 'Q1049511' (1554)
Mid: 'Q1186977' (2), 'Q983026' (2)
Tail: 'Q455833' (1), 'Q3283342' (1), 'Q17183770' (1), 'Q7491877' (1), 'Q30308127' (1)

---

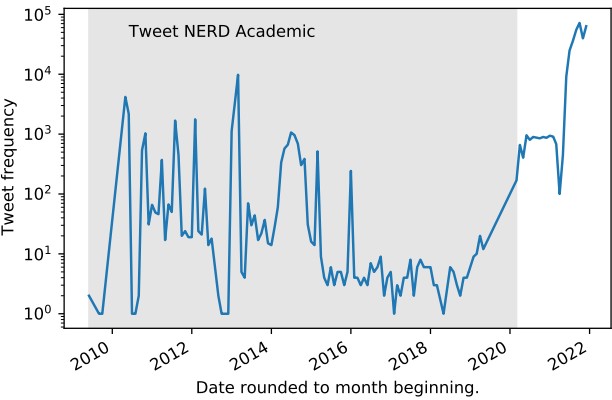

Figure 3: Temporal distribution of Tweets in `TweetNERD`. `TweetNERD-Academic` shaded Grey.

**Temporal Distribution of Dataset.** `TweetNERD` consists of 340K Tweets sampled non-uniformly across 12 years from 2010 to 2021. This includes a smaller but temporally diverse subset of Tweets from existing academic benchmarks, re-annotated using our guidelines. Without the academic benchmarks, the dataset consists of 310K Tweets between 2020-01 and 2021-12.

## 4.4 Inter-annotator agreement

**Limitations of current inter-annotator agreement measures for NERD tasks** All Tweets in `TweetNERD` are annotated by three annotators. For classification tasks Cohen's Kappa [McHugh, 2012] is considerd a standard measure of inter-annotator agreement (IAA). However, for NERD tasks, Kappa is not the most relevant measure, as noted in multiple studies (Hripcsak and Rothschild [2005], Grouin et al. [2011]). The main issue with Kappa is its requirement of negative classes which is not known for NER and NERD tasks. Furthermore, NERD task involves a sequence of words or in our case offsets in text making the number of items variable for each text. A workaround is to use Kappa at token level. However, this results in additional issues. First, annotations are done at the Tweet level instead of token level and for our task tokens will depend on the tokenizer used. Second, token level annotaiton lead to an abundance of "O" tags for NER which will overwhelm the kappa statistics. In Derczynski et al. [2016] the evaluation is done using F1 measure between annotations of two annotators. This is reasonable when we have a fixed set of annotators doing annotation on all the Tweets. However, this is not possible for `TweetNERD` as the annotations were collected from a crowd sourcing system where different set of annotators may annotate different Tweets. Hence, the only approach for calculating agreement in our case is agreement among annotators.

**`TweetNERD` NERD agreement** We compute inter-annotator agreement at **mention $m$ and mention-entity** $(m, eid)$ levels. 69% mentions have a majority ($\geq 2$) agreement, of which 38% have agreement from all three annotators. 17% of mention-entities have 100% agreement across all three annotators, 41% have majority ($\geq 2$), and 59% have only single annotator. 40% mention-entities in `TweetNERD-OOD` and 57% in `TweetNERD-Academic` have majority agreement. If we consolidate `AMBIGOUS` and `NOT FOUND` $eid$ as `NIL` the majority agreement goes up to 47%. At the Tweet level, 30% Tweets have majority agreement across all annotated mention-entities. These agreement scores highlight the difficulty and ambiguity of the end to end entity linking annotation task as described in Section 3. While it is possible to resolve some of these ambiguities using a heuristic, we release the dataset in its current format to encourage research in annotation consolidation and evaluation using these annotations. Although, we use majority agreement on mention-entities as our gold dataset for all evaluations described later, our released dataset contains non-majority annotations to enable additional research in this domain.

## 4.5 `TweetNERD` Data Format

We release `TweetNERD` in a non-tokenized format. `TweetNERD` consists of only Tweet Ids and our annotations as suggested by the Public Twitter API[3]. Each `TweetNERD` file consists of Tweet ids, start and end offsets, mention phrase, linked entity, and annotator agreement score (see Figure 4). We provide details in Appendix A on how to convert this format into token label format suitable for training and evaluating NER systems. All mentions are untyped.

| Id | Start | End | Mention | Entity | Score |
|----|-------|-----|---------|--------|-------|
| 1 | 7 | 14 | Twitter | Q918 | 3 |
| 2 | 0 | 5 | Paris | Q90 | 3 |
| 3 | 0 | 4 | Anil | AMB. | 2 |

Figure 4: **Data Format.** Sample Tweets from Figure 2 to illustrate the data format.

Table 5: Evaluating `TweetNERD-OOD` and `TweetNERD-Academic` using existing systems.

| Model | OOD | Academic |
|---|---|---|
| Spacy | 0.377 | 0.454 |
| StanzaNLP | 0.421 | 0.503 |
| SocialMediaIE | 0.153 | 0.245 |
| BERTweet WNUT17 | 0.278 | 0.46 |
| TwitterNER | 0.424 | 0.522 |
| AllenNLP | 0.454 | 0.552 |

(a) NER `strong_mention_match` F1 scores.

| Model | entity_match | | strong_all_match | |
|---|---|---|---|---|
| | OOD | Academic | OOD | Academic |
| GENRE | 0.469 | 0.636 | 0.39 | 0.624 |
| REL | 0.463 | 0.614 | 0.387 | 0.56 |
| Lookup | 0.621 | 0.645 | 0.584 | 0.617 |

(b) Entity Linking given true spans (EL) F1 scores.

| Model | entity_match | | strong_all_match | |
|---|---|---|---|---|
| | OOD | Academic | OOD | Academic |
| DBpedia | 0.292 | 0.399 | 0.231 | 0.347 |
| NLAI | 0.522 | 0.568 | 0.313 | 0.494 |
| TAGME | 0.402 | 0.431 | 0.293 | 0.381 |
| REL | 0.344 | 0.484 | 0.27 | 0.444 |
| GENRE[4] | 0.307 | 0.458 | 0.223 | 0.379 |

(c) End to End Entity Linking (End2End) F1 scores.

# 5 Evaluation on `TweetNERD`

We use `neleval`[5] library for evaluating various publicly available systems on `TweetNERD`. For our evaluations we always map `NOT FOUND` and `AMBIGUOUS` to `NIL`. We describe the metrics and the evaluation setup below for the three NERD tasks: Named Entity Recognition (NER), Entity Linking with True Spans (EL), and End to End Entity Linking (End2End).

**Metrics**   We first describe the main metrics from `neleval` that are used for evaluation across the three sub-tasks defined above. `strong_mention_match` is a micro-averaged evaluation of entity mentions that is used for the NER task. This metric requires a start and end offset to be returned for the mention. For systems that don't provide offsets we infer the offset in the original text by finding the first mention of the identified mention text. `strong_all_match` is a micro-averaged link evaluation of all mention-entities whereas `entity_match` is a micro-averaged Tweet-level set of entities measure. For EL and End2End tasks, we use `strong_all_match` and `entity_match` as evaluation metrics. `entity_match` is more robust to offset mismatches whereas `strong_all_match` requires a strict match. We report F1 scores for each metric described above. F1 is a harmonic mean of precision and recall. Please see Appendix B for details.

## 5.1 Performance of Existing Entity Linking Systems.

In this section we benchmark existing systems for NERD tasks using `TweetNERD` and suggests these as baseline for future evaluations. We also identify a strong heuristic baseline using exact match lookup. All experiments were run on a machine with NVIDIA A100 GPU and 32 GB RAM. We choose our baselines based on the availability of NER, EL, and End2End systems favoring those which are either widely used in literature or tailored for social media datasets.

**Named Entity Recognition.**   For NER we use StanzaNLP [Qi et al., 2020], Spacy, AllenNLP [Peters et al., 2017], BERTweet [Nguyen et al., 2020] fine-tuned on WNUT17 [Derczynski et al., 2017], Twitter NER [Mishra and Diesner, 2016], and Social Media IE [Mishra, 2019, 2020a,b]. We

---

[3] https://developer.twitter.com/en/docs/twitter-api

[4] Using GENRE end-to-end entity linking model for Table 5-c and entity disambiguation model for Table 5-b. Evaluation scores are after removing a few Tweets from the gold set for which the GENRE model fails. Not removing these Tweets and simply returning Null for GENRE only makes a difference in the third decimal point.

[5] https://neleval.readthedocs.io/

chose these for their popularity and relevance for social media data. See more details about the systems in Appendix Section D.1. Table 5a shows that TwitterNER and AllenNLP perform the best on OOD and Academic dataset. We also find that many of the errors of other systems come from incorrect mention start and end offset prediction even when the mention string is correctly identified.

**Entity Linking given True Spans (EL).** For EL we use GENRE (Generative ENtity REtrieval) [Cao et al., 2021], REL (Radboud Entity Linker) [van Hulst et al., 2020], and Lookup. Lookup is a heuristic based system, which given true mentions, predicts most popular entity based on mention candidate co-occurrence in Wikipedia. See details in Appendix Section D.2. Table 5b shows that Lookup is the best overall, while REL and GENRE come close in performance on Academic subset.

**End to End Entity Linking (End2End).** For End2End we use GENRE (Generative ENtity REtrieval) [Cao et al., 2021], REL (Radboud Entity Linker) [van Hulst et al., 2020], TagMe [Ferragina and Scaiella, 2012], DBPedia Spotlight [Daiber et al., 2013], Natural Language AI (NLAI) API from Google. See details in Appendix Section D.3. Table 5c shows that NLAI is the best overall, whereas REL and GENRE come close in performance on Academic subset.

## 6 Limitations

`TweetNERD` is the largest dataset for NERD tasks on Tweets. However, we highlight a few limitations. First, this is a non-static dataset since some of the Tweets referenced by Tweet IDs may become inaccessible at a later date. The inclusion of `TweetNERD-Academic` aims to mitigate this issue as Tweets in that subset have survived a longer duration. Second, the difficulty of our annotation task limits the performance ceiling on `TweetNERD` as highlighted in the inter-annotator agreement section. However, this provides an opportunity to develop systems on such challenging benchmarks. Finally, the offset based format of `TweetNERD` makes it challenging to be benchmarked by traditional NER systems which often rely on pre-tokenized text. Our suggestion for using `neleval` may help address that issue but will require systems to return offsets corresponding to the original text in `TweetNERD` which may be challenging for traditional systems. The `entity_match` eval score is tokenization and offset agnostic but is only applicable for the end to end NERD task.

## 7 Conclusion

We described the largest dataset for NERD tasks on Tweets called `TweetNERD` and performed benchmarking on popular NERD systems on its two subsets `TweetNERD-OOD` and `TweetNERD-Academic`. We hope that the release of this large-scale dataset enables research community to revisit and conduct further research into the problem of entity linking on social media. `TweetNERD` should foster research and development of robust NERD models for social media which exhibit generalization across domains and time periods. `TweetNERD` is available at: https://doi.org/10.5281/zenodo.6617192 under Creative Commons Attribution 4.0 International (CC BY 4.0) license [Mishra et al., 2022]. Check out more details at https://github.com/twitter-research/TweetNERD.

**Acknowledgments and Disclosure of Funding**

We would like to thank Twitter's Human Computation team, specifically Iuliia Rivera, and Marge Oreta for their efforts in designing and setting up the annotation tasks and training the annotators which was instrumental in generating `TweetNERD` data. We would also like to extend our gratitude to the annotators who contributed to this task directly.

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
