# OpenReview forum: "TweetNERD - End to End Entity Linking Benchmark for Tweets"
_NeurIPS.cc/2022/Track/Datasets_and_Benchmarks — NeurIPS 2022 Datasets and Benchmarks _

### Official Review · Reviewer_Tsie · 2022-07-06
**A carefully prepared dataset, but is unlikely a benchmark.**

**Rating:** 7
**Confidence:** 5

**Strengths:**

The dataset is of large scale and the authors have considered many factors in constructing this dataset, from tweets sampling to the annotation guideline, and then to the construction of two subsets.
The authors conduct evaluations of a few models for three problems, NER, entity linking, and end-to-end entity linking.
The authors choose to release raw annotations in addition to the golden labels. The raw annotation enables studies of other research problems on top of the entity linking problems.

**Weaknesses:**

1. The main weakness is the access to the data is limited by Twitter API terms and conditions. I do understand that all users of the dataset have to follow the rules, but the result is different versions of the dataset being used in different papers. As a result, model accuracies are not directly comparable.

2. The authors do not consider dynamics of Wikipedia, which is always being edited. A specific version or dump shall be specified, to avoid minimize the differences used by different authors. Some entity linking models rely on inter-linking between wiki entries from Wikipedia, which is not included in the annotation.

3. The context used during annotation (Lines 50 - 53) might not well match with what is recorded in the dataset. In other words, a model may not be able to access the full context for making the correct linking. Example is the temporal context of the tweet.

4. The paper is in general not well organized and with lots of small issues in writing. For example, "annotation setup" and "inter-annotator agreement" shall be presented together to offer a complete picture of the annotation process.

The authors have well addressed these comments.

**Additional Feedback:**

Please see the weakness and the detailed comments on clarity.

**Clarity:**

The paper is mostly well written but can be better organized as I mentioned earlier in the weakness. Some detailed comments on writing:
1. The use of reference seems not correct. Table, Figure, Sections are not always capitalized in the text.
2. Lines 109-112. In this example, the 95th Academy Award can be determined by the general temporal context, not necessarily mentioned in the tweets?
3. Table 3, do you have a specific definition for entity, phrase, and phrase-entity?

**Correctness:**

The dataset is constructed in a reasonable manner. The authors indeed discuss all considerations on the sampling and annotation details. Yet, there are some points can be made clearer. Are all 340K tweets annotated by the same 3 annotators? (Line 119). If not, how many annotators are there for these 340K tweets and on average how many tweets each annotator annotates?

In Table 4, results for two subsets are reported. It is more meaningful to report the results on the entire datasets. We will have a better understanding of model accuracy on this large dataset, and at the same time, we will know the difficulty levels of OOD and Academic with reference to the entire dataset.

**Documentation:**

The documentation is provided through a URL. The dataset does not provide tweet content, which can only be obtained through Twitter APIs, with the risk of some tweets being deleted.

**Ethics:**

I do not see any ethical concerns here.

**Relation To Prior Work:**

The paper well discusses the existing datasets. In terms of the evaluation, the models are not comprehensive, in Table 4. Nevertheless, this is not a key focus of this paper.

**Summary And Contributions:**

In this paper, the authors present a large-scale dataset for entity recognition and linking for tweets. In terms of size, the dataset is much larger than existing datasets for the same problem settings. Based on the description, the dataset is properly annotated with clear instructions to the annotators. In this sense, the quality of annotations is high, and the authors release raw annotations in addition to the golden labels. The dataset also comes with two carefully constructed subsets targeting easy and challenging cases. The main weakness comes with the limited access and dynamics of the dataset. Following Twitter policy, only Twitter ID is released instead of the raw content. Downloading the dataset at a different time may get a different subset. This would make the results from different papers not comparable. I would then do not consider this dataset to be really a benchmark. The authors also do not mention the dynamics of Wikipedia, being edited every day.

---

> ### Author Response · Authors · 2022-08-29
> **We thank the reviewer for their feedback and comments. We have addressed the weaknesses and general comments below.**
>
> >"The main weakness is the access to the data is limited by Twitter API terms and conditions. I do understand that all users of the dataset have to follow the rules, but the result is different versions of the dataset being used in different papers. As a result, model accuracies are not directly comparable."
>
> This is a genuine concern but is a limitation of any dataset based on social media,  as it is derived from user provided data which is available under Twitter’s terms of service. Hence we suggest evaluation on the tweets are available at the time of evaluation. This approach will require re-running all the comparison systems to perform a correct evaluation against benchmark systems. However, given the availability of significant proportion of tweets (around 50-60) in Academic split (see Table 3) many years after their creation, we can be hopeful that a large portion of this dataset will remain stable across time.
>
> >“The authors do not consider dynamics of Wikipedia, which is always being edited. A specific version or dump shall be specified, to avoid minimize the differences used by different authors. Some entity linking models rely on inter-linking between wiki entries from Wikipedia, which is not included in the annotation.”
>
> Knowledge Bases are dynamic and new entities are added with time and since NERD datasets are not updated with time there might be discrepancies in model evaluation with reference to a static NERD test set. This is a common limitation of Entity linking evaluation. In TweetNERD this would only affect the NIL predictions as opposed to linking predictions. An entity which in 2014 may be marked as NIL (because of absence from Wikidata) may be marked correctly now. This can be addressed easily by factoring in the creation date of the entity in the Wikidata. This way any entity whose creation date in Wikidata is after the Tweet Date can be marked as NIL. This can allow for temporal evaluation. We have updated the paper with this explanation.
>
>
> >“The context used during annotation (Lines 50 - 53) might not well match with what is recorded in the dataset. In other words, a model may not be able to access the full context for making the correct linking. Example is the temporal context of the tweet.”
>
> Our aim is to create the most accurate linking dataset and therefore we encourage the annotators to always select the most accurate entity using contextual information (which is not provided to them via the annotation UI but they use Twitter platform to look for that information). Also, this contextual information is accessible through Twitter API if it’s needed. Therefore, when models are trained on this dataset, they can mention whether they collected and used this contextual information or not.
>
> >“The paper is in general not well organized and with lots of small issues in writing. For example, "annotation setup" and "inter-annotator agreement" shall be presented together to offer a complete picture of the annotation process.”
>
> Thank you for the feedback. We have restructured the paper and created a separate section called “Annotation Setup” which includes all details about the annotation guidelines, the process, a sample figure to illustrate the setup, and the difficulty of the task. We have made “Inter-annotator agreement” as a separate subsection under Section 4 since it warrants discussion over the individual data splits.
>
> > “The use of reference seems not correct. Table, Figure, Sections are not always capitalized in the text.
>
> Thank you for reporting this. We have addressed this in the revised paper.
>
> > Table 3, do you have a specific definition for entity, phrase, and phrase-entity?”
>
> We have added details in the caption of Table 3 in the revised paper.

---

### Official Review · Reviewer_Fioh · 2022-07-19
**Review for paper "TweetNERD - End to End Entity Linking Benchmark for Tweets"**

**Rating:** 5
**Confidence:** 3
**Correctness:** The models are only evaluated based o…

**Strengths:**

The main strengths of the dataset are mentioned above, as contributions: (1) size of the dataset and (2) disagreement-aware dataset.

**Weaknesses:**

Several weaknesses can be identified:
- Is is unclear what makes the proposed dataset better than existing one in terms of temporal bias (the dataset is much larger, but many tweets have been published way before 2020)
- How much have the annotations changed in the TweetNERD - Academic dataset compared to the original datasets? Why was the reannotation necessary?
- Were all annotators familiar with the task? Were there any instructions on how to deal or manage disagreement?
- Why computing inter-rater reliability as percentages instead of the more reliable ways that use proper IRR metrics (Krippendorff's alpha for instance)? There is very little discussion that acknowledges the reasons for disagreement and how they can impact the models that are trained and evaluated with this dataset.
- The dataset does not seem to be balanced, but only F1 scores are reported. Why is that and why is the F1 score the only needed metric?
- The results are discussed in very little to no detail.

**Additional Feedback:**

See questions in the weaknesses section.

**Clarity:**

The paper is written fairly well, but several parts could be rewritten in a more clear and concise manner. At times, the reader needs to re-read a particular section several times to grasp the meaning.

**Documentation:**

The dataset is publicly available online.

**Ethics:**

No ethical concerns

**Relation To Prior Work:**

Prior work is well referenced. However, the authors could be more precise in the way they selected the baselines and which tweets to re-annotate. Furthermore, there is no clear description of how these tweets were initially annotated, why new annotation guidelines were necessary, how much the ground truth changed with regard to the original dataset, etc.
Further comparisons in terms of inter-rater reliability is desirable.

**Summary And Contributions:**

The paper introduces TweetNERD, which is a large dataset of 340K+ tweets, mostly collected across 2020-2021, and which can be used to benchmark NERD systems. The main contribution of the submission, thus, consists in the size of the dataset. Furthermore, all expert annotations have been kept in the dataset, which means that the dataset allows for further experimentation with ambiguous annotations or different levels of disagreement.

---

> ### Author Response · Authors · 2022-08-29
> **We thank the reviewer for their feedback and comments. We have addressed the weaknesses and general comments below:**
>
> > Is is unclear what makes the proposed dataset better than existing one in terms of temporal bias (the dataset is much larger, but many tweets have been published way before 2020)”
>
> Thanks for the comment, we acknowledge the mis-understanding, only the academic split which consists of around 30K tweets are before 2020. The remaining Tweets are from 2020 onwards.
>
>
> > How much have the annotations changed in the TweetNERD - Academic dataset compared to the original datasets? Why was the reannotation necessary?”
>
> We have added details in Table 4 to compare the annotations in TweetNERD-Academic dataset to the existing annotations. It is important to note that not all the previous datasets are NERD datasets. Another reason for the reannotation is to comply with the new annotation guidelines where each reviewer needs to pass a certification quiz to qualify for annotating tweets. This ensures equally good quality and annotations on the academic dataset as is on the entire dataset.
>
> > Were all annotators familiar with the task? Were there any instructions on how to deal or manage disagreement?”
>
> Yes, all annotators are trained with extensive guidelines where each reviewer needs to pass a certification quiz to qualify for annotating tweets in this dataset. The guidelines included instructions on how to use the search interface provided, how to query for candidates, and how to use the temporal context to pick the right candidate. Several examples were provided to illustrate the right process and annotations.
>
> > Why computing inter-rater reliability as percentages instead of the more reliable ways that use proper IRR metrics (Krippendorff's alpha for instance)? There is very little discussion that acknowledges the reasons for disagreement and how they can impact the models that are trained and evaluated with this dataset.”
>
> In the Interrater annotator agreement section, we have added details on why some of the well-known measures for IAA can not be used. NERD tasks are inherently complex due to the nature of the task - and the fact that it involves detecting mentions and then linking mentions to the right knowledge base entity makes it much harder to report such agreement scores. We believe that the majority agreement on mention and mention-entity level is a good way to illustrate the inter-rater agreement here.
>
> > The results are discussed in very little to no detail.”
>
> The goal of our work here is to benchmark existing popular systems/methods on our dataset and not necessarily provide insights into why some methods perform better than others. This would indeed be a valuable analysis in papers developing new methods using the TweetNERD dataset.

---

### Official Review · Reviewer_8KGq · 2022-07-23
**This paper introduces a large-scale tweet datasets for benchmarking named entity recognition.**

**Rating:** 7
**Confidence:** 4
**Correctness:** The claims made in this paper are cor…
**Clarity:** The paper is well organized and clear…

**Strengths:**

The large-scale tweet dataset with manually annotated entity links enables further research on entity linking on social media. The performance of some recent methods does not outperform some classic methods on three evaluation tasks, which motivates further research for robust NERD models.

**Weaknesses:**

1) The process of obtaining TweetNERD-OOD is not clearly described: "and then perform stratified sampling based on Tweet actions to divide these buckets into sub-buckets". Hence it's difficult to distinguish samples from this subset and those from regular subsets. Therefore, the implications of performance of investigated models on TweetNERD-OOD is not very obvious.

2) Analyses of performance from existing entity linking systems lack the potential reason for failures from some more recent models compared with earlier models. Distribution shift could be one reason: although some recent method shows more powerful in their respective evaluation dataset, they become less powerful when directly evaluating on the newly collected tweet dataset without fine-tuning.

3) In the experiments section, only the two subsets, TweetNERD-OOD and TweetNERD-Academic, are tested. However, the large remaining samples aren't investigated, hence their values are hard to tell: e.g., models training on remaining samples can obtain much better performance on the two evaluation subsets.




**Additional Feedback:**

I have a few questions about the evaluation section:
1) only the two subsets are utilized for evaluation. How about the other samples (occupying the majority of the dataset)? Similar performance if we test existing models well-trained on their original datasets? Improved performance when training models on remaining subsets and testing on the two evaluation subsets?

2) As mentioned in the paper, samples from the academic subset have shown in the earlier released datasets, but are re-annotated with this paper's annotation guidelines. Is there any analysis on the annotation agreement of earlier and new linked entities? Would combinations from both sources can help refine the annotations? Or for the same entity mention, the link has changed due to different understanding along time?



**Documentation:**

The access link and license are listed in the paper.

**Ethics:**

Ethical concerns are not discussed in the paper, but the dataset itself is publically available via tweeter-api, and potential ethical concerns should be considered and addressed in the api's developer's agreements and policies.

**Relation To Prior Work:**

In introduction section, the paper discusses the major difference with existing datasets: 1) large-scale; 2) longer temporal coverage, 2) currently valid since deleted tweets from earlier times are not included

**Summary And Contributions:**

Summary:
This paper manually annotates 340K+ tweets across 2010-2021 to benchmark named entity recognition systems and evaluate existing models on three related tasks: Named Entity Recognition, Entity Linking with True Spans, and End to End Entity Linking.

Contributions:
besides the regular tweets, the paper also provides two subsets: TweetNERD-OOD and TweetNERD-Academic, to assess out-of-domain performance and temporal generalization of NER models, respectively.

---

> ### Author Response · Authors · 2022-08-29
> **We thank the reviewer for their positive feedback and comments**
>
> > “The process of obtaining TweetNERD-OOD is not clearly described: "and then perform stratified sampling based on Tweet actions to divide these buckets into sub-buckets". Hence it's difficult to distinguish samples from this subset and those from regular subsets. Therefore, the implications of performance of investigated models on TweetNERD-OOD is not very obvious.”
>
> We have added more details and structured the paper differently to explain this better.  It is correctly observed that the TweetNERD-OOD dataset is a different sampled dataset as compared to the rest of the dataset; and that helps us in benchmarking our models on a varied set of datasets.
>
> >“In the experiments section, only the two subsets, TweetNERD-OOD and TweetNERD-Academic, are tested. However, the large remaining samples aren't investigated, hence their values are hard to tell: e.g., models training on remaining samples can obtain much better performance on the two evaluation subsets.”
>
> Thanks, we acknowledge this limitation. However, we demonstrate the utility of the remaining dataset using its size which can be later utilized for training. We plan to train models on the remaining datasets of TweetNERD and leave that as future work.
>
> >"Ethical concerns are not discussed in the paper, but the dataset itself is publically available via tweeter-api, and potential ethical concerns should be considered and addressed in the api's developer's agreements and policies. "
>
> Thanks, the dataset is accessible via the public API and hence should not have any ethical issues. We also provide guidelines on the dataset page about how to use the data.
>
> >"Only the two subsets are utilized for evaluation. How about the other samples (occupying the majority of the dataset)? Similar performance if we test existing models well-trained on their original datasets? Improved performance when training models on remaining subsets and testing on the two evaluation subsets?"
>
> Yes, we acknowledge improved performance of models when trained on the remaining dataset of TweetNERD except OOD and Academic. We plan to conduct this assessment as future work.
>
> > Analyses of performance from existing entity linking systems lack the potential reason for failures from some more recent models compared with earlier models. Distribution shift could be one reason: although some recent method shows more powerful in their respective evaluation dataset, they become less powerful when directly evaluating on the newly collected tweet dataset without fine-tuning.
>
> That is a relevant concern however it is applicable to all entity linking setups.
>
> >"As mentioned in the paper, samples from the academic subset have shown in the earlier released datasets, but are re-annotated with this paper's annotation guidelines. Is there any analysis on the annotation agreement of earlier and new linked entities? Would combinations from both sources can help refine the annotations? Or for the same entity mention, the link has changed due to different understanding along time?"
>
> We re-annotated the academic subset as not all the datasets in this subset were created for linking tasks; some of them are for NER or other tasks which require annotating them for linking purposes. Another reason is that by re-annotating these academic datasets we make sure that they are compatible with our annotation guidelines similar to the rest of the data. We have also added detailed explanation of why some of the Inter-rater agreement metrics are not suitable for NERD tasks and explain why we use majority agreement as our main measure. We have added some Tweets in Table 2 to compare the annotations from previous datasets with our newly re-annotated academic dataset to illustrate the differences.

---

### Official Review · Reviewer_y55Q · 2022-07-27
**Review of Submssion ID 353**

**Rating:** 6
**Confidence:** 4
**Clarity:** Yes.

**Strengths:**

1. The annotated dataset is released by Twitter. It's larger and more recent than existing ones.

2. The dataset keeps raw annotation data and is flexible.

3. The limitations of the dataset are discussed.

4. The paper is well-written and easy to read.


**Weaknesses:**

1. The comparison to existing benchmarks needs more discussion.

2. I didn't find any description on how candidate entities were generated.

3. The sampling is not uniform, partly due to creating TweetNERD-Academic.

4. Evaluation was done only on the two subsets of TweetNERD, and only some brief results were reported.

**Additional Feedback:**

1. The authors simply claimed that existing datasets have limited set of Tweets, are temporally biased, or are no longer valid because of deleted Tweets. I think the comparison to existing datasets needs more discussion, so users can understand what is really new here. For example, is the annotation setup is a new and better approach compared to those used in existing benchmarks?

2. Figure 1(b) mentioned candidates. How were generated?

3. The sampling is non-uniform. Most tweets are from 2020 to 2021, while older tweets correspond to the TweetNERD-Academic subset. I guess TweetNERD-Academic and its counterpart may have different distributions.

4. In page 4, the paragraph of "Entropy based sampling", from the description, I only see popularity is modeled in the phrase entropy, but don't understand how you model disambiguation difficulty here. Wikipedia provides disambiguation function if you search for entities with ambiguity, e.g., "Java". So why not use it?

5. In page 4, the paragraph of "TweetNERD-OOD", the description is unclear. Based on my understanding, it seems this is a subset of the high entropy bucket.

6. In the experimental evaluation, it is encouraged to evaluate the methods on the entire TweetNERD rather than subsets.

7. For the experiments on entity linking with true spans, as along as true spans are given, I think this becomes a disambiguation problem and you only need to decide which entity the mention refers to. So the choice of baseline approaches needs more consideration.

**Correctness:**

Are the claims made in the submission correct? Yes.

If the submission is a dataset, it is constructed in a sound way? Partially. The pre-2020 and post-2020 parts may differ in distribution.

If it is a benchmark, are the evaluation methods and experiment design appropriate and performed correctly? Partially. The choice of competitors for some experiments needs more consideration.

**Documentation:**

Yes.

**Ethics:**

I don't find any ethical concerns.

**Relation To Prior Work:**

Some statistics is shown, but more discussions would help explain the novelty.

**Summary And Contributions:**

This paper introduces TweetNERD, a dataset for benchmarking Named Entity Recognition and Disambiguation (NERD) systems on Tweets. This is so far the largest and most temporally diverse open-sourced dataset benchmark for NERD on Tweets. Two subsets of the dataset, TweetNERD-OOD and TweetNERD-Academic are also provided. The former is for assessing out-of-domain performance. The latter consists of Tweets from a collection of existing academic benchmarks that have been re-annotated with the new annotation guidelines. The two subsets were evaluated for named entity Recognition, entity linking with true spans, and end-to-end entity linking tasks. Several NER/EL approaches were compared and the results are briefly reported in the paper.

---

> ### Author Response · Authors · 2022-08-29
> **We thank the reviewer for their feedback and comments. We have addressed the weaknesses and general comments below.**
>
>
> > “The comparison to existing benchmarks needs more discussion.”
>
> Detailed description of all the methods we have used for benchmarking the three NERD tasks are present in Appendix D. For NER model descriptions see Appendix D.1, for Entity Linking given true spans see Appendix D.2 and for end to end entity linking models see Appendix D.3.
>
>
> > “I didn't find any description on how candidate entities were generated.” and “Figure 1(b) mentioned candidates. How were generated?”
>
> We have added details about how the candidate entities were generated in the caption of Figure 2 (They use the functionality provided in the interface to query the eligible knowledge base candidates). The candidates are not provided to the annotators, they search for the candidates for a given mention using the Wikidata search interface.
>
>
> > “The sampling is not uniform, partly due to creating TweetNERD-Academic.”
>
> Creating entity linking datasets is a very expensive task as unconditioned sampling has a low hit rate (the ratio of samples that are linked with at least one entity to the total number of samples). Therefore, collecting enough positive samples (samples with linked entities) is time consuming and costly. Sampling from academic datasets (Appendix C contains details about these datasets) helps us alleviate this problem since these datasets are from NER and linking tasks. In addition, having TweetNERD-Academic will be useful for benchmarking and comparison with previous work. In addition, we re-annotated these datasets because not all these datasets are created for linking tasks; some of them are for NER or other tasks which require annotating them for linking purposes. Another reason is that by re-annotating these academic datasets we make sure that they are compatible with our annotation guidelines similar to the rest of the data.
>
> > “Evaluation was done only on the two subsets of TweetNERD, and only some brief results were reported.”
>
> Yes this is by design of the dataset as we have marked OOD and Academic as the recommended splits of the dataset. This is aligned with how the recommended splits are provided in the broad dataset. We encourage others to use the remaining tweets in the dataset for training, development, and in-domain test sets. In future, we plan to benchmark numbers by training linking models specifically trained on the remaining splits of the datasets and developing a Tweet specific model for NERD based on TweetNERD, however the focus of this paper to merely describe this dataset and demonstrate how existing systems perform on this dataset.
>
> > “Some statistics is shown, but more discussions would help explain the novelty.”
>
> We have added comparison to the old datasets in Table 3 as well as comparison with our overlap with Yodie dataset. Found tweets (%) column indicate the % of tweets that can be queried using the Twitter Public API. We have also added Table 2 which is a comparison of TweetNERD-Academic dataset with the existing benchmarks. The idea is to prove how the annotations have changed for tweets that were part of the old datasets.
>
>
>
> > “The authors simply claimed that existing datasets have limited set of Tweets, are temporally biased, or are no longer valid because of deleted Tweets. I think the comparison to existing datasets needs more discussion, so users can understand what is really new here. For example, is the annotation setup is a new and better approach compared to those used in existing benchmarks?”
>
> We have addressed this in Introduction section, section 4.2 lines 138-155, as well as in section C of appendix. We have also included qualitative comparison examples against TGX and Yodie in table 2.

---

> > ### Author Response · Authors · 2022-08-29
> > **Continuation**
> >
> > For the experiments on entity linking with true spans, as along as true spans are given, I think this becomes a disambiguation problem and you only need to decide which entity the mention refers to. So the choice of baseline approaches needs more consideration.
> > We completely agree that in case of true spans the problem becomes a disambiguation problem. The baselines we used like GENRE have two versions, one end2end and one disambiguation model. Therefore, Table 4-b and 4-c use different versions of the GENRE model. We have elaborated on it in the 5.1 and in the appendix section D.
> >
> > > The sampling is non-uniform. Most tweets are from 2020 to 2021, while older tweets correspond to the TweetNERD-Academic subset. I guess TweetNERD-Academic and its counterpart may have different distributions.
> >
> > Yes this is by design. TweetNERD-Academic contains tweets before 2020 to allow assessment across older time-periods as well as comparison of systems build using older data. TweetNERD-OOD and the rest of TweetNERD has tweets from 2020 onwards. We also provided a canonical split of the dataset into training, test, and dev splits based on the remaining tweets in TweetNERD (excluding OOD and Academic), see details in section 4.2 lines 156-168.
> >
> > > In page 4, the paragraph of "Entropy based sampling", from the description, I only see popularity is modeled in the phrase entropy, but don't understand how you model disambiguation difficulty here. Wikipedia provides disambiguation function if you search for entities with ambiguity, e.g., "Java". So why not use it?
> >
> > We utilize entropy using the wikipedia disambiguation as well as anchor text link distribution from DBPedia dumps.
> >
> > > In page 4, the paragraph of "TweetNERD-OOD", the description is unclear. Based on my understanding, it seems this is a subset of the high entropy bucket.
> >
> > Thank you for the feedback. TweetNERD-OOD is not a subset of the high entropy bucket. We have created two separate sections - “Sampling” and “Data Splits'' to avoid this confusion. The data split section has been appended with additional information about the two data splits as well.
> >
> > > In the experimental evaluation, it is encouraged to evaluate the methods on the entire TweetNERD rather than subsets.
> >
> > We encourage evaluation on OOD and Academic. We have also provided canonical splits into training, test, and dev using the rest of the dataset.
> >
> > > For the experiments on entity linking with true spans, as along as true spans are given, I think this becomes a disambiguation problem and you only need to decide which entity the mention refers to. So the choice of baseline approaches needs more consideration.
> >
> > We have separately benchmarked three NERD tasks, 1. Named Entity Recognition, 2. Entity linking with true spans (Disambiguation) and 3. End to End Entity Linking. This is because TweetNERD can be used for all of these tasks. We have benchmarked six methods for task 1, three methods for task 2 and five methods for task 3.

---

### Official Review · Reviewer_6yJa · 2022-07-31
**A large-scale, tweets-based benchmark designed for Named Entity Recognition and Disambiguation (NERD) systems**

**Rating:** 8
**Confidence:** 4

**Strengths:**

1. The proposed dataset is considered a large-scale dataset that can adapt to a wide range of experimental settings.
2. The paper introduces the experimental settings for three different tasks related to entity linking.
3. Compared to existing datasets, the TweetNERD benchmark has notable advantages.
4. The authors performed a set of experiments with recent models. The results indicate the benchmark is challenging for future research.
5. The authors utilize entropy-based sampling and create TweetNERD-ood and TweetNERD academia to improve the diversity of the datasets.

**Weaknesses:**

1. This paper has a limited description of existing work/ datasets in this domain.
2. The paper can be improved with more detailed descriptions of models, methods, metric selections, insights on results, etc. This would make the paper easier to follow.
3. The present of figures and tables, especially for Figure 1, can be improved.


**Additional Feedback:**

N.A.

**Clarity:**

The general flow is good. However, the writing and presentation can be improved. The paper can be better organized for more details.

**Correctness:**

Yes. However, the evaluation methods, metrics, settings, and experiment designs may be further described.

**Documentation:**

Yes.

**Ethics:**

N.A.

**Relation To Prior Work:**

The paper presents a comparison to existing work in Figure 1 but provides limited information on previous work.

**Summary And Contributions:**

This manuscript introduces a new benchmark named TweetNERD for NERD evaluation. The proposed dataset is based on over 340k tweets from a 10+ years time range and is publicly available with the proper licenses.

The authors compare TweetNERD with four existing benchmarks from multiple aspects, including the number of unique entities, entities, mentions, and the number of tweets. The proposed dataset shows remarkable improvement over others. In this paper, the authors demonstrate the experimental settings of TweetNERD for NERD-related tasks. They are Named Entity Recognition (NER), Entity Linking with True Spans (EL), and End to End Entity Linking (End2End). The experimental results with recent models are present in this paper as well. The numbers indicate the benchmark is challenging for current solutions, and there's much to explore for this dataset.

---

> ### Author Response · Authors · 2022-08-29
> **We thank the reviewer for their feedback and comments. We have addressed the weaknesses and general comments below:**
>
>
> > This paper has a limited description of existing work/ datasets in this domain.
>
> We acknowledge this limitation and have addressed it in a new version of the paper. We have included a related works section 1.1 as well as a detailed description of each dataset we considered in TweetNERD academic in appendix section C.
>
> > The paper can be improved with more detailed descriptions of models, methods, metric selections, insights on results, etc. This would make the paper easier to follow.
>
> We have included the descriptions of models/methods in Appendix because of space limitations. For NER model descriptions see Appendix D.1, for Entity Linking given true spans see Appendix D.2 and for end to end entity linking models see Appendix D.3.
>
> > The presence of figures and tables, especially for Figure 1, can be improved.
>
> Thank you for the feedback on improving Figure1. It has been broken into Figure1, Figure2 and Figure4 in the revised paper. We have also moved these individual figures under their respective sections to enable better understanding and create a cohesive reading flow.
>
> > The evaluation methods, metrics, settings, and experiment designs may be further described.
>
> We have added the description of our metrics and baselines in Sections 5 and 5.1 respectively. For a detailed description of metrics see Appendix B. For detailed descriptions of the baselines for all three evaluation settings see Appendix D.

---

### Author Response · Authors · 2022-08-29
**Thanks for the comments.**

Hi everyone,
Thanks for your detailed comments. We have addressed all the comments for each reviewer (see below). We have updated the paper with the changes addressing reviewer concerns.

Key highlights of our changes are as follows:

1. We provide more detailed literature review
2. We provide more details about each dataset in Academic split and our motivation for its inclusion.
3. We compare against annotations in Academic as well as the number of Tweets from each academic dataset we could retrieve using the Twitter API.
4. We have also included a canonical train, dev, and test split of the data in the data download page. This split is different from OOD and Academic and utilizes the rest of TweetNERD dataset. This can allow for traditional in-domain evaluation of Entity Linking systems on TweetNERD.
5. We address issues around dataset deletion by suggesting the evaluation on only the available tweets in the dataset.
6. We address concerns around inter-annotator agreement and describe why the traditional inter-annotation setups are not applicable for NER and Entity Linking tasks. We refer to relevant literature to address this issue.

---

### Meta-Review · Area_Chair_vvnf · 2022-09-09

**Recommendation:** Accept
**Confidence:** 5

**Metareview:**

The authors release TweetNERD, a collection of 340k tweets from a 10+ years time range with the proper licenses for Named Entity Recognition (NER), Entity Linking with True Spans (EL), and End to End Entity Linking (End2End). All reviewers appreciate that this is a large-scale temporally diverse dataset with many advantages over existing benchmarks, useful for multiple tasks, and experiments show that it is challenging for current models. They also find the paper well-written and easy to read. The main shortcoming they find is that related work and existing benchmarks could be more exhaustively described, and the paper could be more clear in certain areas. Some reviewers are also concerned that since the dataset needs to be downloaded by the Twitter API, it could change over time, resulting in slightly incomparable results across groups.

---

### Decision · Program_Chairs · 2022-09-16

Accept